# Performance of A Statistical-Based Automatic Contrast-to-Noise Ratio Measurement on Images of the ACR CT Phantom

**DOI:** 10.3390/jimaging11060175

**Published:** 2025-05-26

**Authors:** Choirul Anam, Riska Amilia, Ariij Naufal, Heri Sutanto, Wahyu S. Budi, Geoff Dougherty

**Affiliations:** 1Department of Physics, Faculty of Sciences and Mathematics, Diponegoro University, Jl. Prof. Soedarto SH, Tembalang, Semarang 50275, Indonesia; riskamilia0@gmail.com (R.A.); ariij.2019@fisika.fsm.undip.ac.id (A.N.); herisutanto@live.undip.ac.id (H.S.); wahyu.sb@fisika.fsm.undip.ac.id (W.S.B.); 2Department of Applied Physics and Medical Imaging, California State University Channel Islands, Camarillo, CA 93012, USA; geoff.dougherty@csuci.edu

**Keywords:** contrast-to-noise ratio, ACR CT phantom, image quality, noise

## Abstract

This study evaluates the performance of a statistical-based automatic contrast-to-noise ratio (CNR) measurement method on images of the ACR CT phantom under varying imaging parameters. A statistical automatic method for segmenting low-contrast objects and for measuring CNR was recently introduced. The method employs a 25 mm region of interest (ROI), rotated in 2° clockwise steps, to identify the low-contrast object by locating the maximum CT value. The CNR was measured on images acquired with different parameters: tube voltage (80–140 kVp), tube current (80–200 mA), slice thickness (1.25–10 mm), field of view (190–230 mm), and convolution kernel (edge, ultra, lung, bone, chest, standard). The automatic results were compared to manual measurements. The automatic method accurately identified the largest low-contrast object. The CNR values from the automatic and manual methods showed no significant difference (*p* > 0.05). The CNR increased with higher tube voltage and current, and with thinner slice thickness. Chest and standard kernels yielded higher CNRs, while edge, ultra, lung, and bone kernels yielded lower ones. The CNR remained stable with minor FOV changes. The statistical-based automatic method provided accurate and consistent CNR measurements across a range of imaging settings for the ACR CT phantom.

## 1. Introduction

Dose optimization in computed tomography (CT) examinations is important due to two main reasons: the increasing global usage of CT and its relatively high radiation dose [1,2,3]. A critical aspect of dose optimization efforts is the selection of appropriate protocol parameters for a specific clinical CT examination [4,5]. Appropriate imaging protocol parameters are crucial since they relate to the image quality and the received dose by the patient, ensuring that unnecessary radiation is not delivered to the patient to obtain a specific image quality for a specific clinical CT examination [6,7,8,9,10]. These reasons lead to an urgency for a better understanding of selecting the appropriate imaging protocol parameters [4,5].

Appropriate selection of imaging parameters will increase the visibility of a lesion, and is mainly influenced by two parameters i.e., image noise and image contrast [11,12,13]. Image noise is defined as a random fluctuation of CT numbers within an image, which causes a mottled appearance in the image. It is unavoidable in diagnostic imaging, and it is significantly influenced by the selected imaging parameters [14,15]. Increasing the tube voltage, tube current, or slice thickness, will increase the number of X-rays received by the detector [5,16], resulting in a reduction in image noise. In addition, the choice of different reconstruction kernels will result in different image noise levels [17]. Image contrast is not only affected by the object contrast (i.e., different linear coefficient linear attenuations of the objects), but it is also affected by the average photon energy used [5,16]. This means that the choice of appropriate energy or energy spectrum of the photon beam is important to produce an image with good contrast [16]. Commonly, employing low energy X-rays (i.e. produced by a low tube voltage) will increase the contrast of the CT image, to an extent depending on the different tissue’s atomic numbers [18]. 

Thus, detecting any lesion in the CT image is simultaneously determined by both contrast and the image noise level [5]. The ratio between these two parameters is defined as the contrast-to-noise ratio (CNR). The CNR indicates whether a low-contrast lesion can be differentiated within the image or not [19,20]. As the CNR of an image increases, the lesion visibility increases, which is of paramount importance for any diagnosis [21]. In abdominal imaging, for example, a high-CNR image plays an important role for diagnosing hypovascular hepatic metastases [22]. Furthermore, the diagnosis of acute ischemic stroke is strongly influenced by the CNR within images [23]. Therefore, it is crucial to periodically evaluate the CNR value of CT images to ensure diagnostic accuracy. 

Quantitative evaluation of the CNR on CT images is regularly carried out in a quality control (QC) program [24]. The CNR measurement is usually performed using specialized phantoms having a low-contrast module, such as the AAPM CT phantom [25], Catphan® phantom [26], and ACR CT phantom [27]. The CNR is evaluated monthly with a tolerance level (based on the ACR standard) of 1.0 [23]. The ACR CT phantom has been widely used in previous studies [27,28,29], but unfortunately, many assessments were performed manually leading to subjective assessments. Automatic assessments of CNR measurements have been introduced by several researchers with various algorithms [19,30,31]. Some automatic methods employed a segmentation method with various addition processes [32,33], and others employed a statistical method [34]. The latter method was recently introduced [34]. It was reported to be superior compared to the segmentation method [34]. However, a previous study on the statistical method did not test the accuracy of the CNR measurements on images with various imaging input parameters [34]. This study, therefore, aimed to evaluate the performance of an automatic method for measuring the CNR based on the statistical method using ACR CT phantom images scanned with various imaging parameters. 

## 2. Materials and Methods

### 2.1. Imaging the Phantom

CNR measurements were performed on images of the ACR CT phantom (Gammex RMI) scanned by a GE Revolution 128 CT scanner (Chicago, IL, USA). Several imaging parameters were varied, i.e., tube voltage, tube current, slice thickness, field of view (FOV) and convolution kernel, as shown in Table 1. For each variation of a parameter, the other parameters were set constant.

### 2.2. Methods of Automatic CNR Measurement

The novel automatic CNR measurement is a statistical-based method. The method utilizes the previously known radial position of the object to place the ROIs precisely at the center of the largest low-contrast object [34].

It is known that the radial position of the largest low-contrast object is 55 mm from the centroid of the object. However, the position of the largest low-object relative to the angle is not exactly known. It is important to note that the phantom should be accurately positioned by experienced medical personnel in the center of the gantry. However, in most cases, some degree of misalignment still exists. To find the largest low-contrast object, a statistical method was used. An ROI of 25 mm in size was located at 55 mm from the center and rotated in 2° clockwise steps (Figure 1). 

At each position of the ROI, the average CT number was calculated. After one complete 360° rotation, a profile of the CT number around one rotation was obtained. The center coordinate of the largest low-contrast object was determined as the position with the maximum CT number. This method determines the position of the largest low-contrast object even if the phantom is not properly located [34]. In addition, an ROI was located at the center of the phantom. From the two ROIs (i.e., one at the image center and at the largest low-contrast object), the CNR value can be calculated using Equation (1).(1)$$CNR=\frac{\bar{{CT}_{object}}-\bar{{CT}_{background}}}{{SD}_{background}}$$

The CNR values from this automatic method were compared with those from the manual method [35]. The manual method involved placing two ROIs at two locations: the center of the phantom and the center of the largest low-contrast object. Subsequently, the contrast, noise, and CNR values were measured. For the manual method, the images were evaluated for a window-width (WW) of 150 and a window-level (WL) of 75. The statistical automatic method was integrated into the IndoQCT software (v25c.14).

### 2.3. Statistical Analysis

The Mann-Whitney U test was performed to investigate the difference between the results of the two methods. This evaluation was performed using IBM SPSS (version 25) with a *p*-value of 0.05. If the result yields a *p*-value higher than 0.05, it means that the results of both methods do not differ significantly.

## 3. Results

### 3.1. Tube Voltage Variation

The algorithm accurately located the ROI within the low-contrast object for the tube voltage variation. Table 2 shows the results of the contrast and noise level measured in the ACR CT phantom images with tube voltage variation. The results indicated that tube voltage had little effect on the contrast in both manual and statistical automatic methods. However, the noise decreased as the tube voltage increased, as expected. The noise decreased from 4.3 HU at 80 kV to 2.7 HU at 140 kV. The contrast and noise measurement results for the two methods were not significantly different.

The results of CNR at various tube voltages for both methods are shown in Figure 2. The highest CNR value was found at a tube voltage of 140 kV, with a CNR of 2.6 for the manual method and 2.8 for the automatic method, while the lowest CNR was found at a tube voltage of 80 kV, with a CNR of around 1.2. Additionally, it can be seen that the automatic method produced slightly greater CNR values than the manual method, with an average difference for both methods at about 4%. Statistical analysis showed that results from both methods did not differ significantly (*p*-value > 0.05).

### 3.2. Tube Current Variation

The algorithm accurately located the ROI within the low-contrast object for tube current variation. Table 3 shows the results of contrast and noise on images scanned with various tube currents. Increasing the tube current had no effect on contrast as expected, with the contrast value remaining constant between 6 and 7 HU. Conversely, as the tube current increased, the noise value decreased. At 80 mA, the noise value was 4.2 HU, which dropped significantly to 2.6 HU at 200 mA. The contrast and noise measurement results for the two methods were not significantly different (*p*-value > 0.05).

The CNR values at various tube currents are shown in Figure 3. The results show that the CNR value tended to increase as the tube current was increased. The CNR value at 80 mA was approximately 1.5 and increased to 2.4 at 200 mA. The difference between the two methods was not significant, while the average difference between both methods was about 5.3%.

### 3.3. Slice Thickness Variation

The algorithm accurately located the ROI within the low-contrast object for all the slice thicknesses used. The results of the contrast and the noise at various slice thicknesses are tabulated in Table 4. The image contrasts were constant as the slice thickness was increased, as expected. Conversely, the measured noise value decreased as the slice thickness increased. The noise value was 5 HU at 1.25 mm and decreased to 2.3 HU at a 9 mm slice thickness. Both methods produced similar results for the contrast and noise level for various slice thicknesses. The contrast and noise measurement results for the two methods were not significantly different for slice thickness variation (*p*-value ≥ 0.05), and the difference between the two methods for measuring contrast was found to be between 0 and 2%. 

The results of CNR measurements for various slice thicknesses are shown in Figure 4. As the slice thickness increased, the CNR value of the image also increased. The CNR value started at 1.2 at a slice thickness of 1.25 mm and increased with increasing slice thickness, reaching a CNR of 3.1 at a slice thickness of 9 mm.

### 3.4. Field of View Variation

The algorithm accurately located the ROI within the low-contrast object for the FOVs used. Table 5 shows the results of contrast and noise measurements at various FOVs. In general, changes in the FOV value used had no significant effect on the resulting contrast and noise values. Additionally, the manual and automatic methods did not differ significantly in measuring these parameters, with the percentage difference ranging from 1 to 3%. 

The CNR measurement results are shown in Figure 5. The results indicated that the FOV used had no effect on the resulting CNR value, which fluctuated in the range of 2 to 3. Additionally, it was found that the average difference between the two methods for measuring the CNR was about 3%.

### 3.5. Kernel Convolution Variation

The algorithm accurately located the ROI within the low-contrast object for the different kernels used. Table 6 shows the results of contrast and noise measurements for various kernels. The kernel used did not significantly affect the contrast value of the image. The lowest contrast value of 6.0 was found with the ultra kernel, while the highest contrast value of 6.9 was found with the bone kernel. The measurement difference between both methods for contrast was in the range of 3 to 15%.

In contrast, the noise value in the images was strongly influenced by the kernel used. The highest noise value of 50 HU was observed with the edge kernel, and the lowest noise value of 2.9 HU was found with the standard kernel, as expected. The measurement difference between the two methods for noise was very low, at approximately 1%.

The CNR results for various convolution kernels are shown in Figure 6. It shows that the CNR value was influenced by the convolution kernel used. The highest CNR value was achieved with the standard kernel, followed by the chest and bone kernels. Conversely, the lowest CNR value was observed with the edge kernel, followed by the ultra and lung kernels. Specifically, the standard kernel produced a CNR of approximately 2.1, while the edge kernel yielded a CNR of 0.1. However, both methods did not differ significantly in measuring the CNR (*p*-value > 0.05). 

## 4. Discussion

Accurate diagnosis in imaging is heavily influenced by the CNR value of the image, as it indicates the ability to distinguish lesions. The CNR value depends on the imaging parameters used, making it crucial to evaluate each parameter and its impact on the CNR value obtained. An automated CNR measurement is preferable to manual measurement, which depends on the subjectivity of the examiner. However, detection of the low-contrast object by automated methods frequently fails due to the difficulty in accurate placement of ROIs within the image using conventional segmentation methods [31,32,33]. To overcome this problem, a method based on a statistical approach was previously introduced [34].

This study evaluated the performance of the statistical method on images obtained with various input parameters, i.e., tube current, tube voltage, slice thickness, field of view (FOV), and convolution kernel. We found that this method is capable of accurately placing ROIs, similar to manual placement, confirming its potential as a reliable alternative for evaluating CNR in medical images. 

The current study found no significant difference between the CNR from the manual and the automated statistical method. The Mann-Whitney U test resulted in a *p*-value greater than 0.05 across all measurement variations. The study corroborates previous research, which tested an automated CNR measurement algorithm using a statistical method on 25 scanners [34]. That study reported that this statistical method was 100% successful in measuring CNR, while conventional segmentation methods had a success rate of only 56%.

Our current study found that there was little impact on contrast (in the range from 5 to 7 HU) with variations in tube voltage. For other parameters, the contrast value tends to remain constant with minimal fluctuation. Although previous studies suggest that increasing tube voltage decreases contrast, this effect is context-dependent and varies based on tissue composition. For tissues such as iodine, using a low tube voltage enhances contrast against surrounding soft tissues [5].

Our results show that the noise value is significantly influenced by the imaging parameters of tube voltage, tube current, and slice thickness, as reported elsewhere [35,36,37,38]. An increase in tube voltage decreased the noise value due to the higher average energy of photons penetrating the object. Similarly, increases in the tube current and slice thickness also reduced noise. This reduction occurs because both parameters result in a higher number of photons reaching the detector [35,36].

The effect of noise on CNR values is evident. A decrease in noise directly causes an increase in CNR, as observed with variations in tube voltage and tube current. Therefore, higher tube voltage and current improve lesion detection, although this comes with an increased radiation dose to the patient [37,38]. Hence, optimization principles should be applied when selecting these parameters.

Selection of slice thickness should be carefully carried out. The decrease in CNR with increasing slice thickness is not the only consideration. It is important to note that thick slices increase partial volume artifacts (PVA), making small lesion detection more difficult [39]. Selecting an optimum slice thickness must be customized according to the specific indication and condition.

Our study found no significant change in contrast and noise values with various FOVs. The noise and contrast values remained relatively stable with minimal fluctuation. This may be due to the narrow FOV range used (190–230 mm). A wider range of FOV variations is needed to better investigate the impact of FOV on CNR.

We confirmed that the choice of reconstruction kernel significantly affects the image noise value. The use of smooth kernels, such as the chest and standard kernels, results in relatively low noise, ranging from 2 to 6 HU. This reduction in noise is due to the averaging effect inherent in smooth kernels [30]. In contrast, sharp kernels such as the edge, ultra, lung, and bone kernels produce much higher noise levels, exceeding 10 HU. Sharp kernels enhance specific areas of interest, such as bone structures, by improving edge definition between regions with large density differences. While this results in sharper images in certain areas, it also leads to increased noise throughout the image due to enhanced spatial resolution.

Our study has several limitations. It was only carried out on a single CT machine with images reconstructed using the filtered back-projection (FBP) method. Evaluations on images obtained with newer reconstruction methods such as iterative reconstruction (IR) or deep learning image reconstruction (DLIR) were not investigated.

## 5. Conclusions

A statistical automatic method for measuring CNR on low-contrast objects with the ACR CT phantom under various input parameters was evaluated. The method was found to accurately locate the ROI within the low-contrast object and produced accurate CNR values. It also found that the value of CNR was highly influenced by the choice of imaging parameters. The appropriate selection of scanning parameters must be considered to produce an image with acceptable CNR levels for lesion detection. Before applying the method to clinical applications, validation tests on different CT scanners should be conducted, as performed by other authoritative studies in the field.

## Figures and Tables

**Figure 1 jimaging-11-00175-f001:**
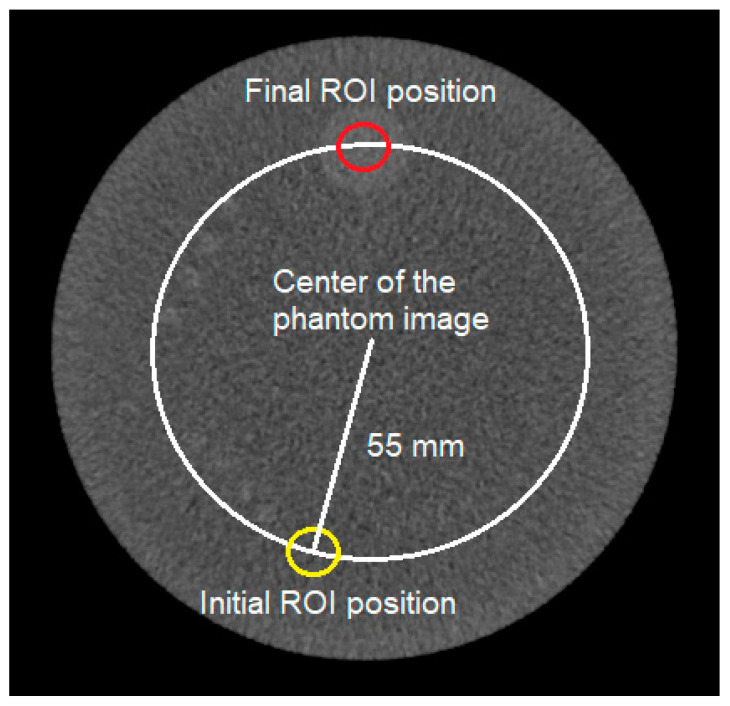
A method to locate an ROI at the largest low-contrast object. The ROI was located at 55 mm from the center and rotated in 2° clockwise steps. The initial ROI can be at any angle. The center coordinate of the largest low-contrast object was determined as the position with the maximum CT number. The yellow ROI indicated the initial ROI position, and the red one indicated the final ROI position.

**Figure 2 jimaging-11-00175-f002:**
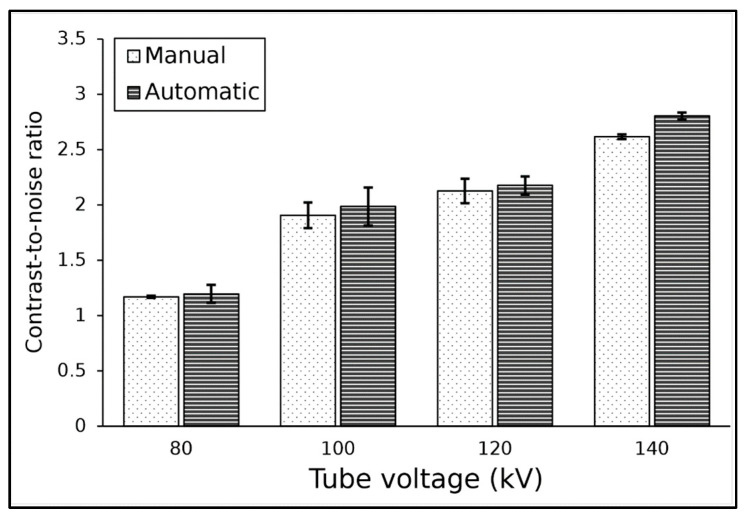
The result of CNR measurement at tube voltages from 80 to 140 kV.

**Figure 3 jimaging-11-00175-f003:**
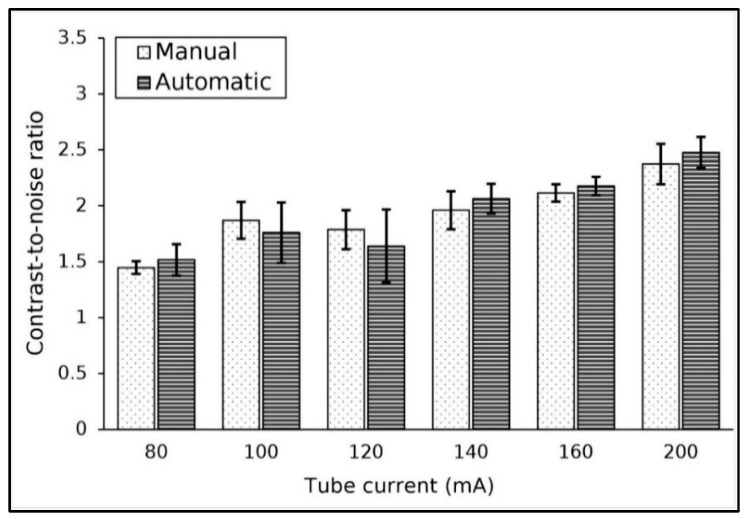
The result of CNR measurement for various tube currents.

**Figure 4 jimaging-11-00175-f004:**
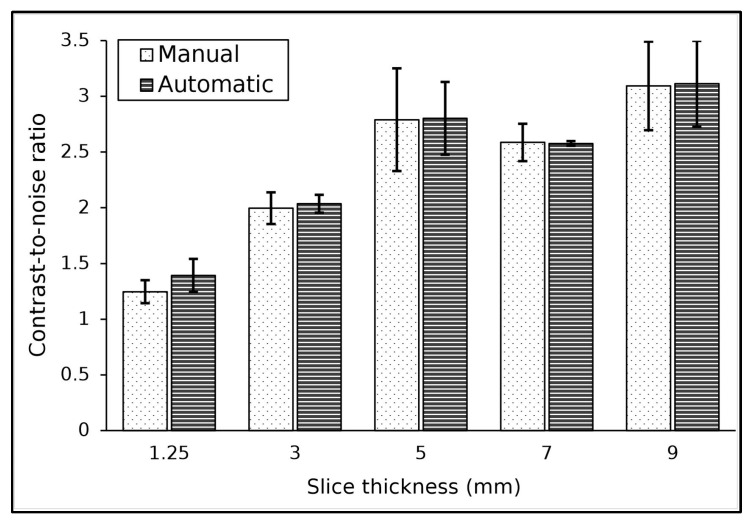
The result of CNR measurement for various slice thicknesses.

**Figure 5 jimaging-11-00175-f005:**
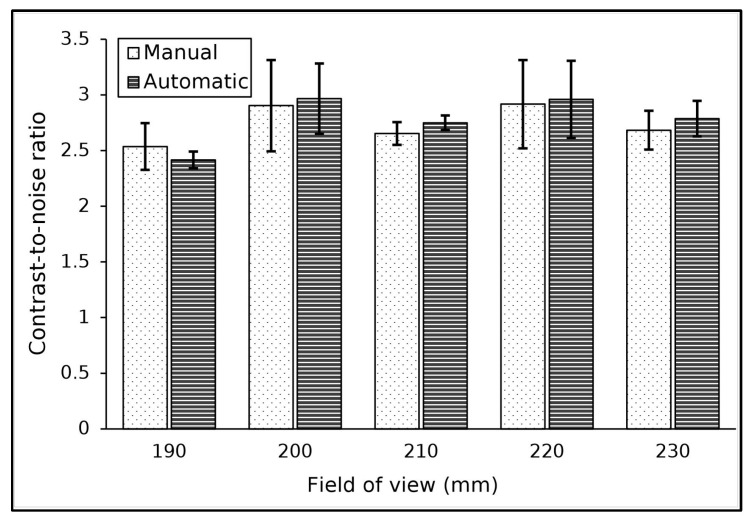
The result of CNR measurements for various FOVs.

**Figure 6 jimaging-11-00175-f006:**
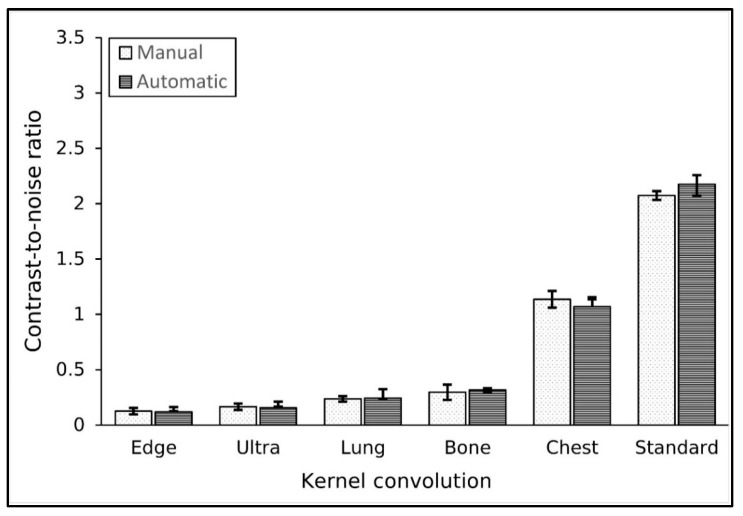
The result of CNR measurement using various convolution kernels.

**Table 1 jimaging-11-00175-t001:** Scanning parameter variations used.

Scanning Parameter	Variation				
	Tube Voltage	Tube Current	Slice Thickness	Field of View	Convolution Kernel
Tube voltage	80, 100, 120, and 140 kV	120 kV	120 kV	120 kV	120 kV
Tube current	160 mA	80, 100, 120, 140, 160, and 200 mA	160 mA	160 mA	160 mA
Slice thickness	1.25 mm	1.25 mm	1.25, 3, 5, 7, and 9 mm	1.25 mm	1.25 mm
Field of view	235 mm	235 mm	235 mm	190, 200, 210, 220, and 235 mm	235 mm
Convolution kernel	Standard	Standard	Standard	Standard	Chest, Bone, Lung, Ultra, Edge, and Standard
Rotation time	0.8 s	0.8 s	0.8 s	0.8 s	0.8 s
Acquisition mode	Spiral	Spiral	Spiral	Spiral	Spiral
Pitch	0.53	0.53	0.53	0.53	0.53

**Table 2 jimaging-11-00175-t002:** The results of contrast and noise measured in images using various tube voltages.

Tube Voltage (kV)	Contrast (HU)				Noise (HU)			
	Manual	Automatic	Difference	*p*-Value	Manual	Automatic	Difference	*p*-Value
80	5.1 ± 2.1	5.1 ± 0.3	0.5%	0.51	4.3 ± 0.1	4.2 ± 0.0	1.2%	0.05
100	6.8 ± 0.5	7.2 ± 0.6	5.7%	0.28	3.6 ± 0.1	3.6 ± 0.1	1.4%	0.05
120	6.4 ± 0.2	6.4 ± 0.1	0.5%	0.83	3.0 ± 0.1	2.9 ± 0.2	1.7%	0.05
140	6.8 ± 0.5	7.3 ± 0.5	7.3%	0.50	2.7 ± 0.1	2.7 ± 0.1	1.3%	0.05

**Table 3 jimaging-11-00175-t003:** The result of contrast and noise measurements using various tube currents.

Tube Current (mA)	Contrast (HU)				Noise (HU)			
	Manual	Automatic	Difference	*p*-Value	Manual	Automatic	Difference	*p*-Value
80	6.2 ± 0.5	6.4 ± 0.6	4.5%	0.51	4.3 ± 0.2	4.3 ± 0.2	0.0%	0.83
100	7.2 ± 0.3	6.7 ± 0.7	6.7%	0.28	3.9 ± 0.2	3.9 ± 0.1	0.6%	0.66
120	6.6 ± 0.5	6.9 ± 0.8	5.0%	0.83	3.7 ± 0.1	4.4 ± 1.4	18.8%	0.83
140	6.2 ± 0.2	6.6 ± 0.4	5.7%	0.28	3.2 ± 0.2	3.2 ± 0.0	0.3%	0.66
160	6.3 ± 0.3	6.4 ± 0.1	1.0%	0.83	3.0 ± 0.1	2.9 ± 0.2	1.9%	0.51
200	6.6 ± 0.3	6.6 ± 0.3	1.0%	0.49	2.7 ± 0.1	2.7 ± 0.1	1.4%	0.38

**Table 4 jimaging-11-00175-t004:** The result of contrast and noise measurements using various slice thicknesses.

Slice Thickness (mm)	Contrast (HU)				Noise (HU)			
	Manual	Automatic	Difference	*p*-Value	Manual	Automatic	Difference	*p*-Value
1.25	6.3 ± 2.1	6.9 ± 2.1	9.0%	0.13	5.1 ± 0.1	5.0 ± 0.2	2.3%	0.66
3	6.6 ± 0.5	6.7 ± 0.4	1.5%	0.83	3.3 ± 0.1	3.2 ± 0.1	4.2%	0.05
5	7.3 ± 0.8	7.4 ± 0.8	1.9%	0.51	2.6 ± 0.1	2.6 ± 0.0	1.0%	0.66
7	6.9 ± 0.5	6.8 ± 0.6	1.5%	0.83	2.7 ± 0.1	2.7 ± 0.1	1.8%	0.51
9	7.1 ± 0.6	7.2 ± 0.6	2.0%	0.51	2.3 ± 0.1	2.3 ± 0.1	0.9%	0.49

**Table 5 jimaging-11-00175-t005:** The result of contrast and noise measurements using various FOVs.

Field of View (mm)	Contrast (HU)				Noise (HU)			
	Manual	Automatic	Difference	*p*-Value	Manual	Automatic	Difference	*p*-Value
190	6.4 ± 0.4	5.9 ± 0.3	6.9%	1.00	2.5 ± 0.1	2.5 ± 0.1	2.4%	0.51
200	7.0 ± 0.7	7.0 ± 0.6	0.6%	1.00	2.4 ± 0.1	2.4 ± 0.1	1.9%	0.66
210	6.6 ± 0.3	6.8 ± 0.1	3.0%	1.00	2.5 ± 0.0	2.5 ± 0.0	0.7%	0.26
220	7.2 ± 0.6	7.2 ± 0.7	0.2%	1.00	2.5 ± 0.2	2.4 ± 0.1	1.4%	0.83
230	6.5 ± 0.2	6.7 ± 0.2	2.7%	1.00	2.4 ± 0.1	2.4 ± 0.1	1.0%	0.83

**Table 6 jimaging-11-00175-t006:** The result of contrast and noise measurements in various convolution kernels.

Kernel Convolution	Contrast (HU)				Noise (HU)			
	Manual	Automatic	Difference	*p*-Value	Manual	Automatic	Difference	*p*-Value
Edge	6.5 ± 1.3	5.9 ± 2.0	4.6%	0.83	50.7 ± 0.9	50.1 ± 1.9	1.0%	0.83
Ultra	6.0 ± 1.1	5.5 ± 2.0	14.7%	0.50	35.3 ± 0.2	35.1 ± 0.6	0.6%	0.51
Lung	6.8 ± 0.6	6.8 ± 2.1	13.4%	0.51	28.8 ± 1.5	28.4 ± 0.7	1.5%	0.66
Bone	6.9 ± 1.6	7.4 ± 0.3	6.9%	0.28	23.4 ± 0.4	23.3 ± 0.6	0.1%	0.83
Chest	6.8 ± 0.7	6.5 ± 0.6	3.5%	0.83	6.1 ± 0.1	6.1 ± 0.1	0.4%	0.51
Standard	6.2 ± 0.3	6.4 ± 0.1	6.4%	0.83	3.0 ± 0.1	2.9 ± 0.2	1.1%	0.83

## Data Availability

Data are available on request from the authors.

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
