# Peer review of "Performance of A Statistical-Based Automatic Contrast-to-Noise Ratio Measurement on Images of the ACR CT Phantom"

_2313-433X, 2025, doi:10.3390/jimaging11060175_

Round 1
Reviewer 1 Report
Comments and Suggestions for Authors
Dear authors, thank you for sharing the manuscript.
Please find below my main recommendations:
Abstract:
Line 12: "segmenting low-contrast object" should be "segmenting low-contrast objects"
Line 13–14: I suggest: “The method employs a 25 mm region of interest (ROI), rotated in 20 clockwise steps, to identify the low-contrast object by locating the maximum CT value.”
Intro:
Good explanation of clinical relevance and background. Justifies the need for improved CNR measurement methods.
Line 31: “a choice of appropriate protocol parameters” consider “the selection of appropriate protocol parameters”
Line 44: “different kernel reconstructions” consider “different reconstruction kernels”
Line 50: “produced by la ow tube voltage” should be “produced by a low tube voltage”
Line 63: “la ow-contrast module” typo, should be “low-contrast module”
Line 66: “the many were manually performed” should be “many assessments were performed manually”
Methods:
Proper presentation of parameters and imaging protocol.
Line 96: “a ROI with size of 25 mm” consider “an ROI of 25 mm in size”
Line 100: “with size was located” remove “with size”
Results:
Proper systematic presentation of experimental variations.
Line 139: “an average difference of about 4%” → clarify that it’s between automatic and manual CNR.
Minor tense inconsistency in multiple places. Use past tense consistently when referring to experimental outcomes (e.g., “was found”, “were measured”).
Discussion:
Interprets results well concerning previous literature. Clear justification of the advantages of the statistical method.
Line 232: “the difficulty in the accurate placement” consider “due to difficulty in accurate placement...”
Line 251: “Although it was previously stated that an increase in tube voltage would decrease the contrast value...” consider rephrasing: “Although previous studies suggest that increasing tube voltage decreases contrast, this effect is context-dependent and varies based on tissue composition.”
Line 273: “to better to investigate” typo; remove one “to”: “to better investigate”
Line 286: “were not investigated .” remove extra space.
Conclusion:
Direct and appropriately cautious about clinical applicability. Reinforces key contributions.
Line 294: “acceptable CNR level to detect the presence of lesion” consider: “acceptable CNR levels for lesion detection”
Author Response
Abstract:
Line 12: "segmenting low-contrast object" should be "segmenting low-contrast objects"
Response: Thank you for the correction. We have corrected it as suggested. Please see the revised manuscript (page 1, lines 11-12).
Line 13–14: I suggest: “The method employs a 25 mm region of interest (ROI), rotated in 20 clockwise steps, to identify the low-contrast object by locating the maximum CT value.”
Response: Thank you so much for your suggestion. We have revised the manuscript accordingly (page 1, lines 13-15).
Intro:
Good explanation of clinical relevance and background. Justifies the need for improved CNR measurement methods.
Line 31: “a choice of appropriate protocol parameters” consider “the selection of appropriate protocol parameters”
Response: Thank you for your suggestion. We have revised it accordingly (page 1, lines 31-32).
Line 44: “different kernel reconstructions” consider “different reconstruction kernels”
Response: Thank you and we have revised it (page 2, line 45).
Line 50: “produced by la ow tube voltage” should be “produced by a low tube voltage”
Response: Thank you. It is typo. We have revised it (page 2, line 50).
Line 63: “la ow-contrast module” typo, should be “low-contrast module”
Response: Thank you. We have revised it (page 2, line 63).
Line 66: “the many were manually performed” should be “many assessments were performed manually”
Response: We have revised it accordingly (page 2, line 66).
Methods:
Proper presentation of parameters and imaging protocol.
Line 96: “a ROI with size of 25 mm” consider “an ROI of 25 mm in size”
Response: We have revised as suggested. Thank you (page 3, line 97).
Line 100: “with size was located” remove “with size”
Response: We have revised into the correct meaning one (page 3, line 100).
Results:
Proper systematic presentation of experimental variations.
Line 139: “an average difference of about 4%” → clarify that it’s between automatic and manual CNR.
Response: Thank you for your comment. We have revised it into “an average difference between both methods was about 4%” to clarify the sentence (page 4, line 140).
Minor tense inconsistency in multiple places. Use past tense consistently when referring to experimental outcomes (e.g., “was found”, “were measured”).
Response: Thank you so much for your suggestions. We have revised the inconsistency and the inappropriate tenses choices throughout the results.
Discussion:
Interprets results well concerning previous literature. Clear justification of the advantages of the statistical method.
Line 233: “the difficulty in the accurate placement” consider “due to difficulty in accurate placement...”
Response: Thank you. We have revised it accordingly (page 9, lines 233-234).
Line 251: “Although it was previously stated that an increase in tube voltage would decrease the contrast value...” consider rephrasing: “Although previous studies suggest that increasing tube voltage decreases contrast, this effect is context-dependent and varies based on tissue composition.”
Response: We have rephrased it as suggested. Please see the revised manuscript (page 9, lines 251-253).
Line 273: “to better to investigate” typo; remove one “to”: “to better investigate”
Response: We have revised it accordingly (page 10, line 274).
Line 286: “were not investigated .” remove extra space.
Response: We have removed the extra space (page10, line 287).
Conclusion:
Direct and appropriately cautious about clinical applicability. Reinforces key contributions.
Line 294: “acceptable CNR level to detect the presence of lesion” consider: “acceptable CNR levels for lesion detection”
Response: Thank you for the suggestion. We have revised it accordingly (page 10, line 295).

Reviewer 2 Report
Comments and Suggestions for Authors
The manuscript is well written, with clear and fluent scientific English.
The data are presented in a structured manner, and both tables and graph are well-designed and easy to interpret. Figure 1 is clear.
It is a highly technical and original article. Its main limitation, in my opinion, is that the study was conducted using only a single scanner. Anyway, from my perspective, there are no major issues to address, but a few minor improvements could enhance the overall quality of the work.
1) The background is well presented; however, I believe that readers would benefit from a more detailed discussion of the main methods for quantitative evaluation of the CNR, particularly those automatic approaches based on segmentation using various additional processing steps. These methods are only briefly mentioned at line 69. Moreover, in the discussion section, it would be appreciated if the author could provide a comparison with other studies, if available, as currently the comparison is limited to only one study (reference 34).
2) In the conclusion section I believe it should be specified that, before any clinical application, validation tests on different CT scanners should be conducted, as performed by other authoritative studies in the field.
3) There are a few minor inconsistencies and typos throughout the manuscript. While these will likely be corrected during the editorial revision process, it is advisable to carefully review the text to ensure consistency in layout, spacing, and referencing (for example, line 63 "la ow-constrast module", line 65 "the many", line 99 "a ROI with size was", line 108 "an ROI").
Author Response
Comments and Suggestions for Authors
The manuscript is well written, with clear and fluent scientific English.
The data are presented in a structured manner, and both tables and graph are well-designed and easy to interpret. Figure 1 is clear.
It is a highly technical and original article. Its main limitation, in my opinion, is that the study was conducted using only a single scanner. Anyway, from my perspective, there are no major issues to address, but a few minor improvements could enhance the overall quality of the work.
- The background is well presented; however, I believe that readers would benefit from a more detailed discussion of the main methods for quantitative evaluation of the CNR, particularly those automatic approaches based on segmentation using various additional processing steps. These methods are only briefly mentioned at line 69. Moreover, in the discussion section, it would be appreciated if the author could provide a comparison with other studies, if available, as currently the comparison is limited to only one study (reference 34).
Response: Statistic detection of the low-contrast object for measuring CNR is new concept. It has just introduced. Currently, it is only one publication for it. This study is to evaluate the method by comparing its results with the manual measurement. We have added this explanation in the revised manuscript (page 2, lines 70-71).
- In the conclusion section I believe it should be specified that, before any clinical application, validation tests on different CT scanners should be conducted, as performed by other authoritative studies in the field.
Response: We have given an additional explanation to specify the test in the conclusion section as the suggestion (page 10, lines 296-297).
- There are a few minor inconsistencies and typos throughout the manuscript. While these will likely be corrected during the editorial revision process, it is advisable to carefully review the text to ensure consistency in layout, spacing, and referencing (for example, line 63 "la ow-constrast module", line 65 "the many", line 99 "a ROI with size was", line 108 "an ROI").
Response: Thank you so much for your comment. We have revised the manuscript thoroughly.

Round 2
Reviewer 1 Report
Comments and Suggestions for Authors
I am satisfied with the revisions made by the authors and have no further comments.
Thank you.
Reviewer 2 Report
Comments and Suggestions for Authors
The revisions made to the manuscript were minimal but effective. They contributed to improving the clarity and the scientific value of the work.